A new species of the archaic primate Zanycteris from the late Paleocene of western Colorado and the phylogenetic position of the family Picrodontidae

Burger Benjamin John benjamin.burger@usu.edu
Department of Geology, Utah State University , Uintah Basin Regional Campus, Vernal, UT , USA
Jungers William
Electronic publication date: 2013 Oct 29
Publication date: 2013
Volume: 1
Electronic Location ID: e191
Received 2013 Jun 12; Accepted 2013 Oct 8
Copyright: © 2013 Burger
Copyright year: 2013
Copyright holder: Burger
License: This is an open access article distributed under the terms of the Creative Commons Attribution License, which permits unrestricted use, distribution, and reproduction in any medium, provided the original author and source are credited.
License URL: https://creativecommons.org/licenses/by/3.0/

Keywords: Primate, Fossil, Teeth, Diet, Paleocene, Picrodontidae, Pleisadapiformes, Mammal

Funding: University of Colorado William H. Burt Fund Colorado Scientific Society Ogden Tweto Fund Geological Society of America Jeff Deen Memorial scholarship Support came from the University of Colorado William H. Burt Fund, Colorado Scientific Society Ogden Tweto Fund, a grant from Shell, a Geological Society of America research grant, and the Jeff Deen Memorial scholarship. The funders had no role in study design, data collection and analysis, decision to publish, or preparation of the manuscript.

==============================
A new species of an archaic primate (Pleisadapiformes) is described based on a maxilla containing the first and second upper molars from the Fort Union Formation, Atwell Gulch Member in northwestern Colorado. The preserved teeth show the unusual dental characteristics of members of the rare and poorly documented Picrodontidae family, including an elongated centrocrista and wide occlusal surface. The new species is placed within the genus Zanycteris (represented by a single specimen from southern Colorado). This placement is based on similarities in regard to the parastyle, curvilinear centrocrista, and wider anterior stylar shelf on the upper molars. However, the new species differs from the only known species of Zanycteris in exhibiting an upper first molar that is 30% larger in area, while retaining a similarly sized upper second molar. Phylogenetic analysis supports the separation of the Picrodontidae family from the Paromomyidae, while still recognizing picrodontids position within Pleisadapiformes. The unusual dental features of the upper molars likely functioned in life as an enhanced shearing surface between the centrocrista and cristid obliqua crests for a specialized diet of fruit. A similar arrangement is found in the living bat Ariteus (Jamaican fig-eating bat), which feeds on fleshy fruit. The new species showcases the rapid diversification of archaic primates shortly after the extinction of the dinosaurs during the Paleocene, and the unusual dental anatomy of picrodontids to exploit new dietary specializations.

Introduction

The family Picrodontidae consists of rare fossil mammals known only in the late Paleocene (Torrejonian and Tiffanian North American Land Mammal Ages (NALMA)) of North America. Upon the initial discovery of the Picrodontidae Zanycteris in 1917, paleontologists placed the fossil within the Order Chiroptera (Matthew, 1917; Simpson, 1935; Simpson, 1937). Indeed, there is a close resemblance between Zanycteris and some of the fruit-eating bats of the New World, such as the living genus Ariteus (Jamaican fig-eating bat). This morphological similarity is exhibited in the upper first molar which is broadly shaped and greatly expanded. The expansion of the occlusal surface of the upper first molar is likely a reflection of similar diet, rather than of any similar phylogenetic relationship to fruit eating bats, since such specializations are absent in early fossil bat lineages (Simmons & Geisler, 1998). More recently studies have positioned the enigmatic Picrodontidae as aberrant members of archaic primates (Szalay, 1968). Researchers have viewed picrodontids as stemming from a Purgatorius-like ancestor (Tomida, 1982), a Palaechthon-like ancestor (Szalay, 1968) or more derived members of the Paromomyoidea (Szalay, 1968; Silcox, 2001; Silcox & Gunnell, 2008).

The relationship to archaic primates is strengthened by the presence of an enlarged incisor in a lower jaw of the picrodontid Picrodus recovered from Swain Quarry of the Fort Union Formation, middle Paleocene (Torrejonian) in Carbon County, Wyoming (Szalay, 1968; Williams, 1985). The enlarged incisor is often considered a synapomorphy of Plesiadapiformes, and is also found in the early Paleocene genus Purgatorius (Clemens, 2004), although this trait is also found in groups outside of Plesiadapiformes, such as the Apatemyidae (West, 1973).

Picrodontids remain very elusive fossil mammals, with only a handful of specimens known from a narrow span of time during the middle to late Paleocene in North America (Silcox & Gunnell, 2008); in fact this is only the second known specimen of the rare genus Zanycteris.

This paper reports on the occurrence of a new species of Zanycteris discovered in the late Paleocene (Tiffanian) deposits of the Fort Union Formation, Atwell Gulch Member in northwestern Colorado, and also discusses the phylogenetic relationship of picrodontids among various groups of archaic primates living in North America during the Paleocene.

Figure 1 Image of fossil specimen.

Buccal and occlusal views of the holotype specimen, a maxilla containing the upper first and second molars. UCM 87378.

Figure 2 Strict consensus tree of the most parsimonious trees generated from the phylogenetic analysis.

Strict consensus tree of the most parsimonious trees generated from the phylogenetic analysis of 113 upper dental characters scored against the 58 North American archaic primates known from the Paleocene. Temporal ranges during the Paleocene are shown for each species by blackened line.

Methods & Materials

The fossil reported in this paper was recovered during geological mapping of the Citadel Plateau Quadrangle during the 1970s. The site has also produced a diverse fauna of mammals from the Fort Union Formation in western Colorado’s Piceance Creek Basin (Burger & Honey, 2008). The site is referred to as University of Colorado Museum (UCM) Locality number 92177. Continued collection at the site over the years has produced additional fossil mammals (Table 1). However, the new species is represented by a single recovered specimen, despite 30 years of sporadic collection at the site.

Table 1 Mammal Faunal List for UCM locality 92177.

Mammalia	
Allotheria	
Multituberculata	
Ptilodus kummae	
Ectypodus musculus	
Theria	
Erinaceomorpha (“Apheliscidae”)	
Haplaletes serior	
Litomylus ishami	
Phenacodaptes sabulosus	
Haplomylus simpsoni	
Lipotyphla	
Leptacodon tener	
Mesonychia	
cf. Sinonyx sp.	
Procreodi	
Thryptacodon australis	
Arctocyonides mumak	
Carnivoramorpha	
Protictis proteus	
Protictis cf. schaffi	
Primates	
Nannodectes gazini	
Plesiadapis fodinatus	
Chiromyoides gigas	
Zanycteris honeyi new species	
Ignacius frugivorus	
Ignacius sp.	
Carpodaptes cygneus	
Condylarthra	
Ectocion medituber	
Phenacodus grangeri	
Phenacodus magnus	
Pholidota	
Propalaeanodon sp.	

In the Piceance Creek Basin the Fort Union Formation is synonymous with the Atwell Gulch Member, which has been included as a member of the Wasatch Formation (Donnell, 1969) or DeBeque Formation (Kihm, 1984). In this paper I refer the Atwell Gulch Member, as the sole member of the Fort Union Formation, as it has been mapped elsewhere within the Piceance Creek Basin in western Colorado (Hail & Pipiringos, 1990; Hail & Smith, 1994).

The Fort Union Formation (Atwell Gulch Mbr.) varies in thickness from 350 m in the northeast to 196 m in the south, and is divided into upper and lower informal units (Hail & Pipiringos, 1990). The fossil site UCM 92177 is 261 m below the upper contact of the Late Paleocene Fort Union Formation (Atwell Gulch Member), with the Early Eocene Wasatch Formation (Molina Member). The lower unit of the Fort Union Formation consists of light-grey to light-brown sandstones; olive, purple, dark-reddish-brown claystone; and mudstones that are highly variegated. Large ribbon and sheet sandstone bodies are common in the north of the basin, where they can form massive sandstone cliffs measuring upwards of 25 m thick, although most are 5–10 m thick. Pebbles are exclusively composed of sedimentary rocks, including claystone and mudstones.

The upper unit of the Fort Union Formation (Atwell Gulch Mbr.) is composed of carbonaceous shales; thin coals; and thin, but persistent highly calcareous sandstones. Mudstones and claystones are less common in the upper unit. Selenite is common, especially in the carbonaceous shales and coals.

Fossil mammals are abundant in the lower unit of the Fort Union Formation, especially in the variegated beds found in association with UCM locality 92177. Fossil invertebrates, such as the bivalve Unio and a variety of gastropods are common in the upper unit, indicating a progression over time toward a more lacustrine environment. Large accumulations of gastropods are common in the upper unit, which have been interpreted as lakeshore accumulations (Hanley, 1974). Order PRIMATES Linnaeus, 1758

Family PICRODONITDAE Simpson, 1937

Genus ZANYCTERIS Matthew, 1917

ZANYCTERIS HONEYI new species

Holotype—UCM 87378 right maxilla with upper first and second molars.

Etymology—Posthumously named in honor of James G. Honey for his discovery of the holotype, type locality and for his kindness in allowing me to study this collection.

Horizon—Fort Union Formation, Atwell Gulch Member, 261 m below the top contact with the Wasatch Formation.

Localities—Only known from UMC locality number 92177.

Diagnosis—Z. honeyi exhibits an anteriorly protruding parastylar lobe on M1/. Differs from Z. paleocenus by having a 30% large M1/ area, while retaining a similar sized M2/ area to Z. paleocenus. Differs from Picrodus in lacking an extended parastyle on the M2/, having a better developed anterior stylar shelf, and postprotocrista on the M1/. Furthermore, the M1/ centrocrista is more curvilinear than Picrodus. Unlike Draconodus, crenulations occur in the trigon basin of M1/.

Description—The holotype (UCM 87378) is the only specimen known from the Piceance Creek Basin (Fig. 1). However, this specimen preserves morphology to indicate that it differs from Z. paleocenus from southwestern Colorado. The enlarged metastyle on the M1/ projects buccally from the posterior edge, and the stylar shelf bulges from the mid-point of the tooth. The paracingulum (the shelf formed by the paraconule) extends anteriorly, yet the stylar shelf encircling the paracone is poorly formed and resembles Picrodus rather than Zanycteris in shape. The strong postprotocrista of UCM 87378 extends directly toward the vestigial metaconule. This direct course of the postprotocrista results in a more enclosed trigon basin that closely resembles Z. paleocenus. The M2/ exhibits a wide stylar shelf, featuring a large parastyle. However, UCM 87378 typifies Zanycteris in lacking the greatly extended parastyle on the M2/ that is found in specimens of Picrodus. Despite wear, the molar paracone and metacone form a W-shaped crest across the midline on the M2/, while the broad protocone expands along the postprotocrista, a typical plesiadapiform trait. The M1/ measures 2.54 mm in length and 2.46 mm in width. The M2/ measures 1.26 mm in length and 1.60 in width.

Comparison—Previous measurements of the holotype of Z. paleocenus (Simpson, 1935; Szalay, 1968) report a length of 2.05–2.20 mm in length and 1.87–2.00 mm in width for the M1/, indicating that the new species (UCM 87378) is both larger and broader. However, reported measurements of the M2/ (1.25–1.30 mm in length and 1.60–1.70 mm in width) are similar in dimensions to UCM 87378. This indicates that while the first molar is enlarged, the second molar is of equal size between the two species of Zanycteris. This enlargement of the first molar likely served a functional role in providing a larger surface area for slicing between the centrocrista and cristid obliqua (Szalay, 1968). In some ways, UCM 87378 resembles Picrodus, such as the reduced anterior stylar shelf on the M1/ (Scott & Fox, 2005). However, other features more closely resemble Z. paleocenus, including the arrangement of the postprotocrista on the M1/ and smaller parastyle on the M2/. These features support inclusion of UCM 87378 within the genus Zanycteris rather than Picrodus. UCM 87378 and the holotype of Z. paleocenus (AMNH 17180) are the only two specimens of Zanycteris currently documented (Simpson, 1937; Scott & Fox, 2005). Zanycteris appears to be restricted to Colorado, while Picrodus has been found in Wyoming, Montana, Alberta and recently New Mexico (Simpson, 1937; Williams, 1985; Silcox & Gunnell, 2008; Scott & Fox, 2005; Silcox & Williamson, 2012).

Phylogenetic Analysis

The acquisition of the highly specialized dentition found within Zanycteris and other members of the Picrodontidae remains a mystery. For example, how quickly did the specialized dentition evolve during the Paleocene? Among the known Plesiadapiformes, which one is most closely related to the family Picrodontidae and could possibly represent the ancestral condition for the specialized dentation exhibited by Zanycteris?

To evaluate these questions and to work toward reconstructing the evolution of the specialized upper dentition of Picrodontidae, a phylogenetic analysis was undertaken using the morphological characteristics of the dentition of known Plesiadapiformes and outgroups (Paradectes, Cimolestes, and Leptacodon), which lived during the Paleocene in North America. The character matrix consisted of 113 dental characters, 97 of which were adopted from Silcox (2001). The analysis included 58 fossil taxa of contemporary North American Paleocene primates. A heuristic search using Mesquite version 2.75 (Maddison & Maddison, 2011) produced 6,579 most parsimonious trees (597 steps, consistency index [CI] = 0.36, retention index [RI] = 0.74). The strict consensus tree shows Zanycteris honeyi as closely related to Zanycteris paleocenus within a monophyletic clade of Picrodontidae (Picrodus, Draconodus, and Zanycteris) (Fig. 2). The family Picrodontidae was found to be within a clade consisting of Plesidapidae and Carpolestidae, rather than a placement within Paromomyoidea (Silcox & Gunnell, 2008). This phylogenetic position postulates that the expansion of the occlusal surface seen in the upper molars of both paromomyids and picrodontids is convergent, having evolved independently during the Paleocene. Possible ancestors of picrodontids are the early Paleocene taxa Plesiolestes, Torrejonia, Phoxomylus, and Talpohenach, while paromomyids appear to have arisen from the early Paleocene with a Palaechthon or Anasazia or the poorly known Premnoides like ancestor. Although there is ambiguity concerning the phylogenetic position of these primitive taxa, there is strong support for monophyletic clades of Paromomyidae, Picrodontidae, Plesiadapidae, and Carpolestidae, as well as a monophyletic clade of Plesiadapiformes, with the addition of Micromomyidae and Microsyopidae. Overall the resulting phylogenetic tree supports a position of Zanycteris honeyi within the Picrodontidae family, and that this new species is closely related to other archaic primates from the Paleocene of North America.

Discussion

The members of the Picrodontidae are exceptional in the development of a dentition that maximizes the shear forces along the long contact between the centrocrista on the upper molar (composed of a tall crest between the paracone and metacone) and the cristid obliqua that spans much of the length of the elongated talonid basin of the lower molar (Szalay, 1968). The molar teeth were thus probably specialized for cutting through hard outer husks of fruits and nuts. This was accomplished by positioning the fruit along the outer (buccal) shearing surface, which was greatly expanded (anteriorly and posteriorly) to maximize the amount of contact, much like a pair of long sharp scissors (Shaw, 1917). A similar expansion of the centrocrista is found in the upper molars of the Jamaican fig-eating bat (Ariteus), which feeds on the native Jamaican naseberry also known as sapodilla (Manilkara zapota), a fruit with a fleshy but firm texture (Sherwin & Gannon, 2005). Thus in both Ariteus and Zanycteris the major shearing surface is between the crests of the centrocrista above and the cristid obliqua below, demonstrating a similar specialized diet on fruit. This arrangement differs substantially from paromomyids, which retain distinct paracone and metacone cusps on the upper molars, with no development of a long and tall centrocrista between the two cusps. Rather, paromomyids expand the upper molars by broadening the postcingulum to form a wide talon basin on the lingual edge of the tooth. This broadening of the tooth functioned to expand the surface area particularly for holding food during mastication (Shaw, 1917). Thus paromomyids, such as Phenacolemur, broaden the upper molars to allow increased surface area for a larger and a more varied diet, while picrodontids, such as Zanycteris, expanded the upper molars to increase the shearing surface for a more specialized diet of a particular style of fleshy fruit. Further research on dental specialization within these groups might reveal why picrodontids have a limited stratigraphic range (Torrejonian to Tiffanian), when compared to the closely related, but dentally distinct paromomyids which ranged from the early Paleocene (Puercan) until the late Eocene (Duchesnean) (Silcox & Gunnell, 2008).

Conclusions

In summary, the new species Zanycteris honeyi typifies the unique characteristics that set apart the Picrodontidae from other archaic primates known from the Paleocene of North America. Phylogenetic analysis supports the separation of the Picrodontidae family from the Paromomyidae family, while still recognizing their position within Pleisadapiformes. Further fossil discoveries, particularly cranial and postcranial remains will likely enable more confident placement of this unusual group of archaic primates among the evolutional tree during this pivotal time of primate diversification shortly after the extinction of the dinosaurs.

Supplemental Information

Supplemental Information 1 Supplemental Character List

List of characters used in phylogenetic analysis.

Click here for additional data file.

Supplemental Information 2 NEXUS File Used in Phylogenetic Analysis

Click here for additional data file.

Special thanks to Jaelyn Eberle at the University of Colorado at Boulder for her support in my graduate studies which included this research. I would like to thank my committee Herbert H. Covert, Mary Kraus, Matthew Pranter, (University of Colorado) and Henry Fricke (Colorado College); Toni Culver, the collection manager at the University of Colorado Museum, as well as various members of the field crew including Alex Dutchak, Karen Lloyd, Pat Monaco, Andrea R. Bair, Ian J. Sweeney, Lou Taylor and Lea Ann Jolley. Fieldwork was conducted under BLM permit #C-60170 issued to the University of Colorado Museum.

Additional Information and Declarations

Competing Interests

Author Contributions

Field Study Permissions

New Species Registration

The author declares there are no competing interests.

Benjamin John Burger conceived and designed the experiments, performed the experiments, analyzed the data, contributed reagents/materials/analysis tools, wrote the paper.

The following information was supplied relating to ethical approvals (i.e., approving body and any reference numbers):

Fieldwork was conducted under the United States Bureau of Land Management (BLM) permit #C-60170 issued to the University of Colorado Museum.

The following information was supplied regarding the registration of a newly described species:

Zootaxa

urn:lsid:zoobank.org:act:AAE7738F-A4FC-4BF4-A0FF-E9FE35F38EBE.

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
