# Peer review of "A new species of the archaic primate Zanycteris from the late Paleocene of western Colorado and the phylogenetic position of the family Picrodontidae"

_PeerJ, doi:10.7717/peerj.191_

## Round 0.1 · original submission · Major Revisions

Two reviewers see real merit in your paper on these rare fossil primates, but both request much better photographs (especially since you are creating a new type specimen). Both reviewers also raise substantive issues with your phylogenetic analysis that need to be addressed (and which may require new analyses). Additional recommendations are mostly constructive, and attention to these suggestions will serve to improve the diagnosis and the discussion of its significance for primate paleobiology. Careful editing is needed to eliminate various typos and spelling errors.

·

Basic reporting

This paper describes a new species of Zanycteris from the Paleocene of Colorado....it is well written and well organized and I had only a few minor comments which I detail below. In general I find no problem with publishing this paper once the minor fixes are made.

Experimental design

Not applicable

Validity of the findings

There seems no doubt that Z. honeyi is a new species of Zanycteris - I found the ecological analysis a little wanting - I agree that the teeth of Z. honeyi are of a frugivorous animal, similar to Ariteus (often mis-spelled in the mansucript as Artiteus) - but I'm not sure why thet "restricts the stratographic range" of picrodontids to the middle and late Paleocene only - surely there were plenty of fruits available in the Eocene as well....all of those real primates must have been eating something in the Eocene

Additional comments

I find this paper to be more or less good as it is...there are few places where it needs a little editing or rewording....

The only major complaint I have is that Figure 1 showing the holotype specimen really is unacceptable for publication - it is blurry and out of focus - the buccal view is nearly impossible to interpret at all - in this day and age of microCT scanners and microscopic imaging with stacking software surely much better images than these can be provided. Also, it looks like there is still adhering matrix on the occlusal surfaces of the molars - this should be cleaned off beofre re-imaging.

Also I have a list of otrher comments:

Abstract - Ariteus is mis-spelled in fourth line from bottom

Methods and Material section - "diverse fauna of fossil mammals" is cited in line 39 and repeated again in line 42 citing Table 1 (I didn't see Table 1 anywhere) - no need to repeat this same information twice

Comparisons of Z. honeyi - lines 104-105 the authors notes differnces with Picrodus - he should cite the Scott and Fox, 2005 publication here

Discussion - line 146 - I think I'd be a little less dogmatic here and say that the "molar teeth were probably specialized for cutting...."

Line 15 - Ariteus mis-spelled again

Line 155 - paracone and metacone cusps (delete the "s" after paracone and metacone)

Lines 162-165 - I see no basis for saying this - yes paromomyids and picrodontids are different dentally but no actual analysis has been done to substantiate the claim that picrodontids were limited to the middle and late Paleocene by their dental specializations - very few plesiadapiforms made it past the end of the Paleocene in general - I think I'd just drop these lines.

I'm not entirely sure why yet another phylogenetic analysis is being run here - the strict consensus tree differs quiet a lot from other, more rigorous trees - there are so many polytomy's that I'm not sure I believe any of what is being depicted....also the tooth images on the tree are so small as to be unrecognizable as teeth - probably should just eliminate them

That's all I have

·

Basic reporting

Picrodontids are incredibly rare primates, so this paper is certainly worthy of publication. However, there are a few significant issues with the paper that need to be addressed before it can be published. Most serious, the argument for the attribution to Zanycteris is not clearly made. As discussed by both Silcox and Gunnell (2008) and Scott and Fox (2005) the key diagnostic difference between this genus and Picrodus is an anteriorly protruding parastylar lobe in the former, and yet the author suggests that the new specimen belongs in Zanycteris on the basis of a “shorten [sic] parastyle”. Looking at the image, I think that the specimen is actually Zanycteris and does have an anteriorly protruding parastylar lobe, but this certainly isn’t clear in the discussion. What’s more, the image of the specimen is very poor, which makes it difficult to determine if the description as presented is correct. For example, I really can’t tell if the M2 is complete or broken buccally—this is critically important since the relative size of M1 vs. M2 is the key diagnostic trait for the new species. In the buccal view it is impossible to tell tooth from bone—I actually think it would be okay to just cut this figure. For upper teeth, a buccal view isn’t really needed. In the occlusal view it is clear that the specimen has been terribly over-whitened, so that there is a thick layer of whitening material coating the whole specimen, with beaded up material obscuring a good part of the morphology. My guess is that moisture either on the specimen, or in the bulb of the whitening tube, caused the powder to bead up. The image is also somewhat out of focus. Simply put, Figure 1 is not publishable as is. The specimen needs to be cleaned, and a much lighter layer of whitening used. Alternatively, an SEM or microCT image could be substituted.

A few more minor details:

Abstract, line 5: should be “BY a single specimen”
Abstract line 6: should be “shortenED parastyle”
Abstract line 9: should be “similarLY sized upper second molar”
Abstract line 12: should be cristid obliqua, not cristid oblique

Line 20: “more recently studies…” requires a reference –the key paper here is Szalay 1968
Line 25: should be “is strengthenED by the…”
Line 28: “The enlarged incisor is considered a synapomorphy of Plesiadapiformes…” Clemens did suggest this, but Plesiadapiformes has never come out as a monophyletic group (in a holophyletic sense, i.e., excluding euprimates or Dermoptera) in any cladistic analysis. And it doesn’t come out as demonstrably monophyletic in your own results—Purgatorius is part of an unresolved polytomy with the rest of the plesiadapiforms and Plagiomenidae, Apatemyidae etc. For this reason, it is inappropriate to talk about this character as a synapomorphy, at least without a qualification.
Line 31: refs are needed for the time span of picrodontids (i.e., after “in North America”)
Line 69: It is standard to include several more layers in the systematic paleontology listing (e.g., at a minimum Order and Family)
Line 94: It is unclear what “the broad protocone spreads into the postprotocrista” means. Clarify.
Lines 163, 164: when used informally, taxonomic names should not be capitalized (e.g., should be paromomyids and picrodontids)
Figure 2: “Elpidophorus” and “stonleyi” are spelled incorrectly

Experimental design

With respect to the phylogenetic analysis, it seems strange to me to limit the sample to Paleocene mammals. Because of the vagaries of fossil preservation, it is not necessarily the case that younger forms are irrelevant in assessing the ancestry of older forms. This is particularly an issue with respect to the discussion about the possible ancestors of different plesiadapiform groups (lines 133-135). Without including Premnoides douglassi, the putative ancestor of Paromomyidae (see Gunnell, 1989), it really isn’t appropriate to discuss the possible descendants of the mentioned palaechthonids. I’m not necessarily suggesting that the author needs to expand his sample, since the systematic questions he is asking here are fairly limited in scope. However, he should resist the temptation to over-interpret the results.

There are also a number of issues of clarity with the character list that need to be addressed:

Character 1: not clear what is meant by a “winged postcingulum”—this is not standard dental terminology. Please define and clarify.

Character 4: how close to they need to be in size for it to be “nearly”? Define more precisely.

Characters 5, 6, 12: These characters are essentially written as double negatives (i.e., the “present” state implies the absence of something), which is confusing, and non-standard. Re-write (e.g., “Postprotocrista on M2 Present = 0, Absent = 1)

Character 7: “reduced” implies an evolutionary direction. Define based on observed morphology (e.g., short)

Characters 8, 13, 14: It is inappropriate and imprecise to make reference to a particular genus in the definition of a character. Describe what’s there.

Character 10: It is a bad idea to combine two potentially independent elements of morphology in a single character (e.g., size and position in this case) since you have no way of knowing a priori if they always covary.

Character 14: groove in what position?

Character 112: Not clear what is meant by “small” and “enlarged” here. Enlarged relative to what?

Validity of the findings

There is an inaccuracy in how the results are reported. On lines 127-128 the author states that “The strict consensus tree shows Zanycteris honeyi as the sister-taxa [sic] to Zanycteris paleocenus…” No, it doesn’t—the two taxa form a polytomy with Picrodus. The nature of their relationships is unresolved on the tree as presented. This must be clarified. Also, “taxa” is plural—it would be the sister taxON.

Additional comments

I as glad to see that you named this for Jim Honey. He was a very nice man--I was sorry to hear of his passing.

---

## Round 0.2 · Minor Revisions

If theses are the best photographs the author can produce, then I think they are adequate if not "perfect" -- as the re-review notes.

Unless you disagree, in your final version please make reference to "the diagnostic trait for this genus."
Please try to clarify or modify text with respect to
"flanged" and "reduced" (small, smaller, smallest?)

Please correct the statement about your characters since some lower dental features are part of the phylogenetic analysis. I urge you to reflect upon (and perhaps comment on) your decision to limit your characters to those observed only on Palaeocene species.

·

Basic reporting

The author has submitted somewhat improved versions of the images of the teeth. I think it is up to the editor to determine whether these are adequate. It is now possible to see most of the relevant morphology, so my personal assessment is that they are borderline publishable. However, I would not chose to publish an image that has residue partially coating the occlusal surface of the tooth.

Experimental design

I'm experiencing some confusion about the phylogenetic analysis. In his rebuttal letter, the author states that " The character matrix is composed of only upper dental traits", but this is clearly not true (e.g. characters 19-22 all pertain to the lower i1). In any case, it would be a terrible idea to so restrict an analysis. Authors sometimes do this on the mistaken belief that only characters that can be coded for a fossil will affect that fossil's position in a tree. But the common practice of molecular backboning shows why this idea is fallacious. Even though no fossil can be coded for molecular characters, by influencing the relationships of all the OTHER taxa in an analysis, those data will end up influencing the position of the fossil in question. This is also true for morphological data, both with respect to character partitions AND taxon sampling, which is why I think it makes no sense to restrict the analysis to species alive in the Paleocene. However, the goals of this paper are limited in scope, and the results of the analysis are not unreasonable, so this could meet the standard of publishable on those bases.

The character list is generally improved over the preceding revision, although a few characters are still not clear:

ch. 1: adding "flanged" does not help, since this is also not standard anatomical terminology.
ch. 10: the author has once again used "reduced", which implies an evolutionary direction, and as such is inappropriate

Validity of the findings

I am convinced that the species is new and pertains to Zanycteris. I'm puzzled as to why the author continues not to make reference to the diagnostic trait for this genus (i.e., anteriorly protruding parastylar lobe on M1). Surely it wouldn't be difficult to add that in? Or does he disagree?

---

## Round 0.3 · accepted · Accept

Thank you for your cooperation in the review process. I think the reviewer suggestions and your corresponding emendations have served to improve your study substantially. The diversity of North American archaic primates is fascinating.